# Pathways Activated by Infected and Bystander Chondrocytes in Response to Ross River Virus Infection

**DOI:** 10.3390/v15010136

**Published:** 2022-12-31

**Authors:** Elisa X. Y. Lim, Julie A. Webster, Penny A. Rudd, Lara J. Herrero

**Affiliations:** Institute for Glycomics, Gold Coast Campus, Griffith University, Southport, QLD 4222, Australia

**Keywords:** cartilage, alphavirus, extracellular matrix, arthritis

## Abstract

Old world alphaviruses, such as Ross River virus (RRV), cause debilitating arthralgia during acute and chronic stages of the disease. RRV-induced cartilage degradation has been implicated as a cause of joint pain felt by RRV patients. Chondrocytes are a major cell type of cartilage and are involved in the production and maintenance of the cartilage matrix. It is thought that these cells may play a vital role in RRV disease pathogenesis. In this study, we used RNA-sequencing (RNA-Seq) to examine the transcriptomes of RRV-infected and bystander chondrocytes in the same environment. RRV containing green fluorescent protein (GFP) allowed for the separation of RRV-infected (GFP+) and bystander uninfected cells (GFP−). We found that whereas GFP+ and GFP− populations commonly presented similar gene expression profiles during infection, there were also unique signatures. For example, *RIMS2* and *FOXJ1* were unique to GFP+ cells, whilst *Aim2* and *CCL8* were only found in bystander chondrocytes. This indicates that careful selection of potential therapeutic targets is important to minimise adverse effects to the neighbouring uninfected cell populations. Our study serves as a resource to provide more information about the pathways and responses elicited by RRV in cells which are both infected and stimulated because of neighbouring infected cells.

## 1. Introduction

Among the mosquito-borne Old-World alphaviruses, Ross River virus (RRV) is endemic to Australia, Papua New Guinea and the Pacific Islands and can cause debilitating disease in humans. Macropod marsupials serve as one of the animal reservoirs with *Aedes sp*. and *Culex sp.* mosquitoes being the vectors for virus transmission [1,2]. Approximately 5000 human cases are reported annually while sporadic outbreaks have also occurred, such as in 2015 (9544 cases) and 2017 (6915 cases), according to the National Notifiable Diseases Surveillance System (Australia). Most patients typically experience clinical manifestations such as fever, rash, headache, myalgia, fatigue, arthralgia and joint swelling [3]. Chronic joint and muscle pain can persist for several years after the acute infection [3,4]. In some cases, patients were diagnosed with rheumatoid arthritis (RA) after infection by a related alphavirus, chikungunya virus (CHIKV), indicating the importance of studying joint pathology in alphavirus infection [5]. There are no vaccines to prevent RRV disease and symptomatic treatment is commonly provided with use of anti-inflammatory drugs (NSAIDs) which are known to induce adverse effects during long-term use [6]. Recently, pentosan polysulfate sodium (PPS), which is a semisynthetic macromolecular carbohydrate derivative, has shown promise to treat chronic arthralgia [7,8].

Chondrocytes are the main cell type found in the articular cartilage and are important for synthesis and maintenance of the cartilage extracellular matrix (ECM). Chondrocytes are critical for the regulation and expression of pro-inflammatory cytokines, ECM components and enzymes involved in ECM degradation [9]. We have previously shown that in a mouse model of infection, chondrocytes are targets of RRV replication [10]. We also demonstrated, in vitro, that primary human chondrocytes are permissive to RRV infection.

Osteoarthritis (OA) is known to result in joint damage, especially cartilage degradation and inflammation of the synovium. It is also one of the leading musculoskeletal disorders that affect many people worldwide. It has been shown that chondrocyte death is a feature of OA disease formation [11]. However, it remains uncertain if chondrocytes contribute to disease pathogenesis or whether the death of these cells is merely a by-product of cartilage damage that occurs during OA. Recent reports suggest that products formed during ECM breakdown are recognised by the host immune system as damage-associated molecular patterns (DAMPs) and are responsible for the expression of pro-inflammatory cytokines [11,12,13]. Therefore, understanding the responses to RRV infection in chondrocyte cells may help decipher the role these cells play in joint damage.

Despite current extensive understanding of alphaviral inflammatory diseases, there is a critical gap in knowledge regarding how host responses compare when examining both infected and bystander cell populations. To date, very few studies have focused on how viral infections can affect neighbouring uninfected cells [14,15]. Herein we aim to better understand these two cell populations by using RNA-sequencing (RNA-Seq) on primary human chondrocytes. Our approach enables us to distinguish the direct and indirect effects of infection and to compare host cellular pathways elicited during RRV infection in these two cell populations. We know from immunological studies that bystander cells can have an important role in host defence by playing both regulatory and effector roles [16]. A better understanding of how bystander cells respond during infection could lead to novel therapeutic strategies that will help prevent amplifying the consequences of the original infection.

## 2. Materials and Methods

### 2.1. Virus

Virus stocks of the Ross River virus (RRV) T48-GFP (green fluorescence protein) strain were generated with an infectious clone (kindly gifted by Richard Kuhn, Purdue University, West Lafayette, IN, USA) [17]. To generate GFP tagged T48, a second RRV 26S promoter was inserted at the 3′ end of the genome followed by the coding sequence for GFP as previously described [18]. Briefly, in vitro transcription was performed using SP6 RNA polymerase, and the synthesised RNA was electroporated into Vero cells for virus amplification. Infectious virus units were quantified by plaque assay using Vero cells. Vero cells were cultured in OptiMEM supplemented with 3% foetal bovine serum (FBS), and infected cells were maintained with OptiMEM with 1% FBS.

### 2.2. Cells

Human primary chondrocytes (Lonza Bioscience, Sydney, NSW, Australia) were cultured and differentiated according to the manufacturer’s instructions prior to infection. Cells were grown in chondrocyte expansion media: low-glucose DMEM supplemented with 400 μM L-Proline, 100 μM ascorbic acid, 10 mM HEPES, GlutaMAX^TM^, 1× minimum essential medium non-essential amino acids (MEM NEAA), 1× penicillin/streptomycin and 10% FBS. To induce differentiation post-attachment to cell culture surfaces, chondrocyte cells (seeded at 25,000 cells per cm^2^) were maintained in chondrocyte differentiation media: high-glucose DMEM supplemented with 400 μM L-Proline, 100 μM ascorbic acid, 1.25 mg/mL bovine serum albumin (BSA), 0.1 μM dexamethasone, 10 ng/mL transforming growth factor-beta 3 (TGF-β3), 10 mM HEPES, GlutaMAX^TM^, 1× insulin-transferrin-selenium (ITS), 1× MEM NEAA, and 1× penicillin/streptomycin.

### 2.3. RRV-GFP Infection of Differentiated Human Primary Chondrocytes for RNA-Seq

Differentiated human primary chondrocytes (Lonza Bioscience, Sydney, NSW, Australia) were used for all experiments. Chondrocytes were infected with GFP-tagged T48 strain of RRV at a multiplicity of infection (MOI) of 10. At 2 days post-infection, cells were trypsinised, stained with propidium iodide (PI) and resuspended in FACS buffer (5mM EDTA, 25mM HEPES, 1% BSA, 2.5mM MgCl_2_, 50U/mL DNaseI in PBS). Uninfected cells were used as mock controls. After excluding propidium iodide (PI)+ dead cells, PI− live cells were sorted for GFP-positive and GFP-negative using the BD FACS Aria Fusion Cell Sorter flow cytometer (488 nm Blue Laser: 2 PMTs) and downstream analysis was performed using FACSDiva 8.0.1 (BD Biosciences, ANZ, North Ryde, NSW, Australia). Here, GFP-positive cells refer to the population infected with RRV and GFP-negative cells refer to the uninfected (bystander) population.

### 2.4. RNA Extraction and Preparation of cDNA Library

RNA was isolated using the RNeasy Plus Micro kit (Qiagen, Clayton, VIC, Australia). Library preparation and sequencing were performed by the Australian Genome Research Facility (Brisbane, Australia). cDNA libraries were prepared using the Ribo-Zero stranded RNA protocol (Illumina, San Diego, CA, USA). Briefly, RNA samples were ribosomal RNA (rRNA)-depleted, fragmented, and primed with random hexamers for first-strand cDNA synthesis by SuperScript II Reverse Transcriptase (Invitrogen/Thermo Fisher Scientific, Mt Waverley, VIC, Australia). The 2nd strand was synthesised and contained dUTP. Thereafter, cDNA strands were adenylated at 3′ ends and ligated with sequencing adapters. The first strands were amplified through 13 cycles of PCR to form the libraries. Sequencing was performed by Illumina NextSeq (75bp single-end run).

### 2.5. Differential Gene Expression Analysis of RNA-Seq Datasets

Sequence reads were mapped to the Homo sapiens genome (build version HG38) using STAR aligner (v2.5.3a) and transcripts were assembled using StringTie (v1.3.3). Data obtained for the GFP+ and GFP− populations were analysed relative to healthy uninfected cells (mock) from a separate sample. Differential gene expression analysis was performed using edgeR (v 3.24.3) and R (v 3.5.3) with a significance threshold of false discovery rate (FDR) < 0.05 and log2 fold change (log2FC) threshold of 0.58. Count per million (CPM) cut-offs of 50 was used for the GFP+ samples, while CPM cut-offs of 4 were used for GFP− and mock samples. The FDR values are adjusted p-values based on the Benjamini–Hochberg method. The DEGs were sorted by their log2FC values and the top ten up- and down-regulated DEGs for the GFP+ and GFP− populations are listed in Table 1 and Table 2.

### 2.6. Gene Set Enrichment Analysis

Differentially expressed genes were further characterised using BioVenn, Enrichr and AmiGO 2 analysis tools [19,20,21]. The databases used were: Gene Ontology Biological Process (GO2018 BP), Gene Ontology Molecular Function (GO2018 MF), KEGG (KEGG2019) Jensen DISEASES and Human Phenotype Ontology [22,23,24,25,26].

## 3. Results

In this study, overlapping and unique transcriptomic signatures were examined in RRV-infected and uninfected (bystander) primary human chondrocytes. Towards this, cells were infected with RRV-GFP at MOI 10 for 48h (Appendix A). All data analyses were normalised to healthy uninfected cells (mock) from a separate sample. Cellular RNA from the three sample groups: mock, RRV-infected GFP-positive (GFP+) and uninfected bystander GFP-negative (GFP−) were isolated and processed through the RNA-Seq pipeline in triplicate.

Our results showed 5431 and 5352 differentially expressed genes (DEGs) relative to healthy controls in the GFP+ and GFP− sample groups, respectively (Figure 1). When comparing DEG expression in both the GFP+ and GFP− groups, 44.62% were shown to overlap (Figure 2A) whilst 55.38% were unique. The top ten DEGs for the GFP+ and GFP− populations are shown in Table 1 and Table 2.

To further analyse these datasets, we separated the genes into up- and down-regulated DEGs for three categories: 1) genes in the GFP+ group only, 2) overlapping genes which are present in GFP+ and GFP− cells, and 3) genes that are found in the GFP− group only (shown in different shades of green) (Figure 2B).

While we observed differences in gene expression levels of the overlapping genes for the GFP+ and GFP− groups (Table 1 and Table 2), the proportions of up-and downregulated DEGs were similar across all gene profiles (Figure 2B).

We further assessed the DEGs using Enrichr, a platform commonly used for gene set enrichment analysis with many functional annotation libraries such as gene ontology (GO), KEGG and Jensen DISEASES [21]. The top 5 enriched terms for up- and down-regulated DEGs are shown in Figure 3 (GFP+ and GFP−) and Figure 4 (overlapping). We found that DEGs found in GFP+ cells only were significantly enriched for nucleic acid transcription, which was expected for cells which are RRV-infected (Figure 3). On comparison of the GFP+ and GFP− overlapping DEGs, we observed that the upregulated genes belong to processes relating to cytokine-mediated signalling, type I IFN signalling, TNF signalling and negative regulation of viral transcription. This demonstrates that antiviral immune responses were mounted by both groups of cells regardless of infection status (Figure 4). In addition, unsurprisingly, we found that terms relating to arthritis, which is a phenotype of RRV infection, were also overlapping in both cell groups (Figure 4).

To evaluate the role of chondrocytes in disease pathogenesis, we isolated the DEGs related to joint pathology using gene sets from the Human Phenotype Ontology (HPO) database: increased inflammatory response, abnormal joint morphology, and arthritis (Figure 5 and Table 3). There were similar numbers of up- and downregulated DEGs for GFP+ and GFP− groups for the increased inflammatory response and abnormal joint morphology HPO gene sets. We observed a slight increase in the number of DEGs associated with arthritis in the GFP− group compared to GFP+ group (Figure 5C).

As the only cell type present in the cartilage, chondrocytes are responsible for the maintenance of the cartilage through the expression and regulation of ECM components and breakdown molecules. We found that both GFP+ and GFP− groups have similar numbers of DEGs associated with ECM components (≃5 unique genes, respectively) and DEGs involved in ECM assembly (≃6 unique genes, respectively) and disassembly (≃ 10 unique genes, respectively) based on functional annotation gene-sets from the Gene Ontology database (Figure 6). Interestingly, the two populations have many more overlapping ECM genes than unique ones. Altogether, around 110 common genes are involved in these processes. As above, this includes genes that are involved in the structural constituents of the ECM (≃75 genes) as well as ECM assembly (≃18 genes) and disassembly (≃22 genes).

## 4. Discussion

In our study, we investigated the transcriptomic profiles of RRV-infected and uninfected bystander cells to learn more about the direct and indirect effects of RRV on disease pathogenesis in human primary chondrocyte cells. The proportion of DEGs were similar across GFP+ and GFP− cell populations; however, interestingly we did observe some differences in the type of cellular processes modulated by RRV infection (Figure 3 and Figure 4).

As expected, we identified several proinflammatory genes associated with RRV infection. The presence of pro-inflammatory mediators associated with alphavirus infection has been well-characterised [31,37,38]. Many of these cytokines are also associated with RA and have been suggested to be responsible for alphavirus-induced arthritis [31]. Elevated levels of monocyte chemoattractant protein-1 (MCP-1), tumour necrosis factor-alpha (TNF-α), interferon-gamma (IFN-γ), interleukin-6 (IL-6), macrophage migration inhibitory factor (MIF), interleukin-1 beta (IL-1β) and granulocyte–macrophage colony-stimulating factor (GM-CSF) are associated with viral-induced arthritis. Some of these soluble factors have been found in serum and synovial fluids of patients with chronic alphavirus infections [38,39,40,41].

In this study, we have identified several genes related to joint pathology with no known association with alphavirus infection (Table 3). Many of the genes listed in Table 3 encode for ECM structural components such as asporin (ASPN), proteoglycan 4 (PRG4) and tenascin XB (TNXB) were downregulated in both cell populations.

Asporin (encoded by the *ASPN* gene) is a proteoglycan present in the cartilage and binds to collagen fibres to induce mineralisation [42]. While asporin inhibits cartilage differentiation through the blocking of TGF-β/receptor interaction, it has been shown in chondrocytes that TGF-β can induce asporin gene expression [43]. The expression of pro-inflammatory cytokines (such as IL-1β and TNF-α) and chondrocyte dedifferentiation results in reduced *ASPN* expression [43,44]. Asporin is also associated with Toll-like receptor (TLR) signalling, suggesting regulatory roles in inflammatory processes. Overexpression of asporin in periodontal ligaments resulted in dampened TLR2 and TLR4-mediated immune responses through direct binding to these receptors [45]. Activation of TLR signalling initiates a signalling cascade that results in pro-inflammatory cytokine gene expression. Therefore, inhibition of TLR signalling by asporin may result in a reduced inflammatory response. However, the roles of asporin in joint pathology are still not well-elucidated. This proteoglycan has been associated with having protective and risk factors in osteoarthritis (OA) in various studies [42,46]. Our study shows that *ASPN* is down-regulated in both GFP+ and GFP− cells, suggesting that RRV-induced effects on *ASPN* may impact chondrocyte differentiation and contribute to the pro-inflammatory cytokine profile typically observed in RRV infection.

PRG4 is an extracellular matrix (ECM) molecule (encoded by the *PRG4* gene) linked to inflammatory processes and abnormal joint pathologies. This molecule is of particular interest as in our study, we found that it is a key DEG that is associated with all three GO gene sets analysed. However, it has yet been shown to play roles in alphavirus infection (Table 3**)**. This proteoglycan is produced by chondrocytes and synoviocytes, and is found in synovial fluid, where it is a known cushioning and lubricating molecule [47,48]. In recent years, abnormal expression of PRG4 has been associated with joint diseases such as osteoarthritis (OA) and rheumatoid arthritis (RA), and changes in PRG4 expression have been attributed to biological and mechanical stimuli [47].

*TNXB* encodes for a glycoprotein called tenascin XB (TNXB) which is important in the organisation and maintenance of connective tissues. It has been associated with be involved in the synthesis and assembly of collagens. Mutations and/or a deficiency of this gene are associated with a connective tissue disorder called Ehlers-Danlos syndrome [49]. The induction of MMP2 is associated with TNXB deficiency and hypothesised to be caused by the c-Jun N-terminal kinase (JNK) and protein tyrosine kinase (PKK) phosphorylation pathway [50]. The expression of MMPs is associated with cartilage damage in joint diseases as well as alphavirus infections [10,51,52]. However, increased expression of TNXB has been found in synovial cells of OA and rheumatoid arthritis (RA) patients, suggesting the complex and mixed roles of TNXB in joint diseases [53]. In our study, we found downregulation of *TNXB* in both GFP+ and GFP− cells. To explain this observation, we hypothesise that TNXB is involved in the regulation of expression for MMPs during alphavirus disease pathogenesis.

Among the upregulated genes that we identified as potentially involved with RRV joint pathogenesis were *GCH1, FOXJ1, MAGI2*, *SALL4* and *WRAP53*. *GCH1* (GTP cyclohydrolase 1) is a gene that encodes for an enzyme involved in tetrahydrobiopterin (BH4) synthesis and is associated with abnormal joint morphology and arthritis. BH4 is an important cofactor for the synthesis of many biomolecules such as serotonin and nitric oxide. Increased BH4 and nitric oxide synthesis is often induced by inflammatory responses [54]. Upregulation of both GCH1 and BH4 have been associated with pain sensitivity [55]. It has been shown that pain sensitivity returned to normal levels when mice suffering from chronic pain were injected with an inhibitor of GCH1 [56]. While GCH1 has been associated with neurological disorders such as Parkinson’s disease, it has never been linked with alphavirus disease pathogenesis. Our research shows that GCH1 is highly expressed in GFP+ and GFP− cells, indicating that non-infected bystander cells play a role in causing abnormal joint pathology. It is also possible that GCH1 is an early indicator of RRV disease through a heightened state of pain sensitivity which is known to be elevated in RRV patients.

Forkhead box J1 (FOXJ1, gene *FOXJ1*) is a transcription factor important for cilia production expressed on ciliated cells and is also associated with skeletal development. Chondrocytes express non-motile cilia on their cell surface; however, their functions are poorly understood [57]. The chondrocyte cilium carries receptors involved in signalling pathways such as integrins, ion channels, Hedgehog and cAMP. Therefore, it plays an important role in extracellular interactions between cell types and tissues of the joint, though any roles in alphavirus infection are currently unknown. It is hypothesised that the chondrocyte cilium is important for biochemical processes. For example, mechanical loading stimulates Ca^2+^ production only in chondrocytes with cilia. Furthermore, the chondrocyte cilium is also associated with an increased inflammatory response. IL-1β is a pro-inflammatory cytokine with broad activity and induces NF_K_B expression through activation of MyD88 signalling. This, in turn, results in the expression of various pro-inflammatory cytokines such as IL-6, IL-8, TNF-α and members of the IL-1 family, all of which are observed to be expressed during RRV infection [38,39,58,59]. Treatment of chondrocytes with IL-1β resulted in cilia elongation, which affects signalling processes. In addition, it has been shown that the amount and length of cilia increase with severity of OA in cartilage tissue [60]. In our study, we found that the FOXJ1 gene is only significantly up-regulated in the GFP+ cells and we hypothesise that FOXJ1 is involved in RRV-mediated cartilage damage through ciliogenesis.

In our study several interesting genes were upregulated in GFP+ alone including *MAGI2*, *SALL4*, and *WRAP53,* most which had been implicated in human cancers and with no known role in alphaviral infection or disease. *MAGI2* encodes for ‘membrane-associated guanylate kinase, WW and PDZ domain-containing protein which is a member of the MAGI proteins that are known to interact with other scaffold proteins acting to maintain the architecture of cell junctions [61]. MAGI proteins are known to play a role in neuronal stabilisation being highly expressed in the brain with further evidence to suggest a role in tumour suppression [61].

Sal-like protein 4 (SALL4) is a transcription factor in embryonic stem cell self-renewal and pluripotency. SALL4 expression is virtually undetectable in most adult tissues except for germ cells and human blood progenitor cells. In the past decade, research has suggested SALL4 may be a valuable biomarker or therapeutic target for a range of cancers [62].

*WRAP53* encodes for a protein called ‘WD40 encoding RNA antisense to p53′ which has two main components. WRAP53α an RNA regulator that stabilizes p53 RNA and a WRAP53β which is involved in Cajal bodies, telomerase trafficking and DNA repair [63] and has been implicated in primary human cancers.

Transcriptome analysis approaches such as RNA-Seq allow for a quick overview of the cellular processes that are modulated by virus infection. Here, we investigated the gene expression profiles of RRV-infected and bystander chondrocyte cells in the same environment. This analysis allows us to compare the direct and indirect effects of RRV infection and enables us to identify potential targets important in disease pathogenesis for in-depth functional characterisation studies. Validation of selected targets through evaluating protein expression levels and functional studies is important as many pathways play complex roles in gene regulation. Therefore, the observed gene expression profiles may not translate into similar levels of protein produced when assessed through proteomic or functional approaches. Further study into our identified targets would enable for increased understanding of the roles they play during RRV disease pathogenesis.

## Figures and Tables

**Figure 1 viruses-15-00136-f001:**
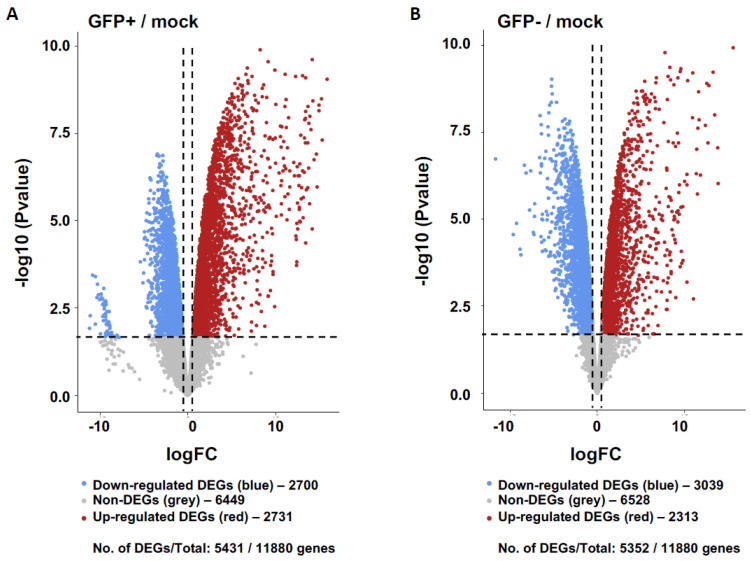
Transcriptomics of GFP+ (RRV-infected) and GFP− (uninfected bystander) populations relative to mock controls of primary human chondrocyte cells. Gene expression of GFP+ cells (**A**) and GFP− cells (**B**) relative to healthy mock controls are shown represented as volcano plots. Differentially expressed genes (DEGs) defined whereby log2 Fold Change (FC) threshold of 0.58 (1.5FC) and False Discovery Rate (FDR) <0.05. The down- and up-regulated DEGs are coloured blue and red, respectively, and non-significant genes are shown in grey.

**Figure 2 viruses-15-00136-f002:**
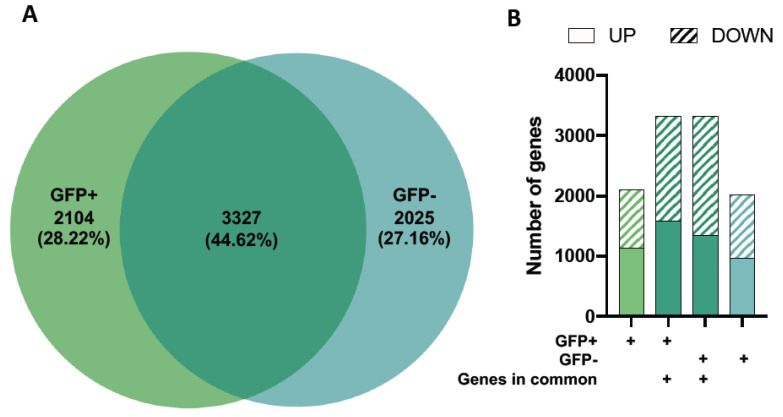
Differential gene expression in GFP+ (RRV-infected) and GFP-mock (uninfected bystander) populations. Differentially expressed genes (DEGs) for GFP+ and GFP− cells relative to healthy mock controls were compared and the proportion of genes expressed as a Venn diagram (**A**). Up- and down-regulated DEGs in each of the four gene profiles (genes in the GFP+ group only, overlapping genes which are present in GFP+ cells, overlapping genes which are present in GFP− cells and genes that are found in the GFP− group only) are shown (**B**).

**Figure 3 viruses-15-00136-f003:**
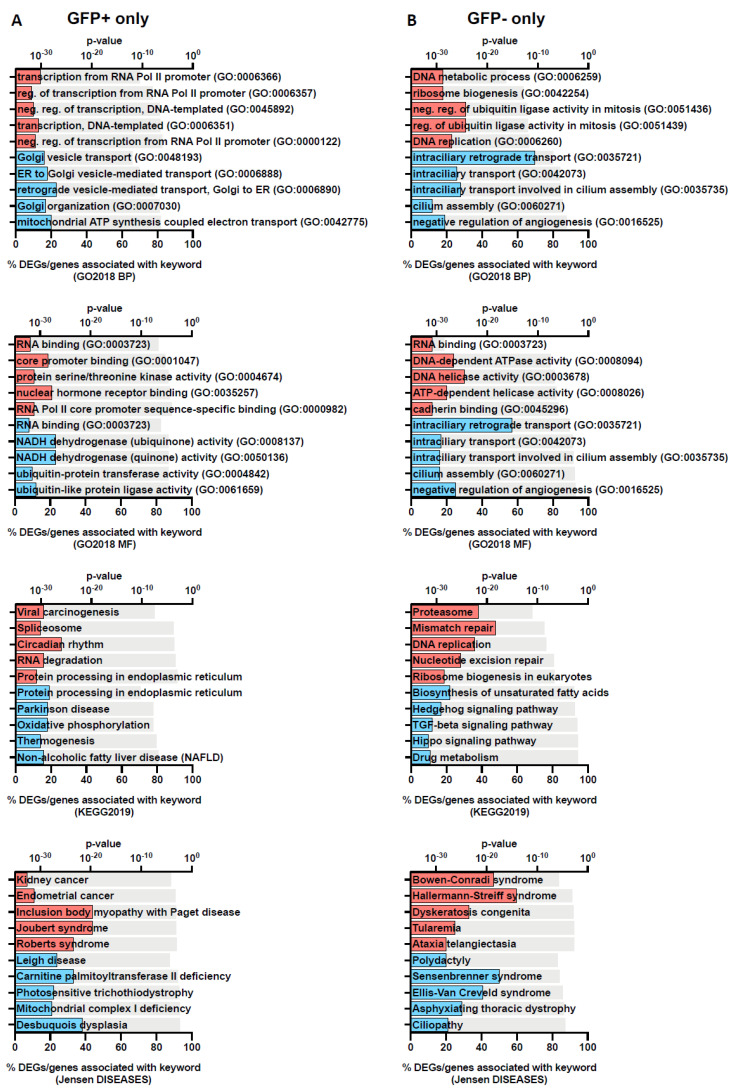
Gene set enrichment analysis of DEGs in GFP+ only and GFP− only populations showing unique genes identified in each population. The top 5 functional annotation gene-set associations for the two gene profiles: GFP+ (**A**) and GFP− (**B**) without overlap were identified using Enrichr with Gene Ontology Biological Process (GO2018 BP) and Molecular Function (GO2018 MF), KEGG and Jensen DISEASES databases. Keywords associated with up- and downregulated genes are coloured red and blue, respectively. The grey bar represents the p-values determined by the Enrichr analysis platform.

**Figure 4 viruses-15-00136-f004:**
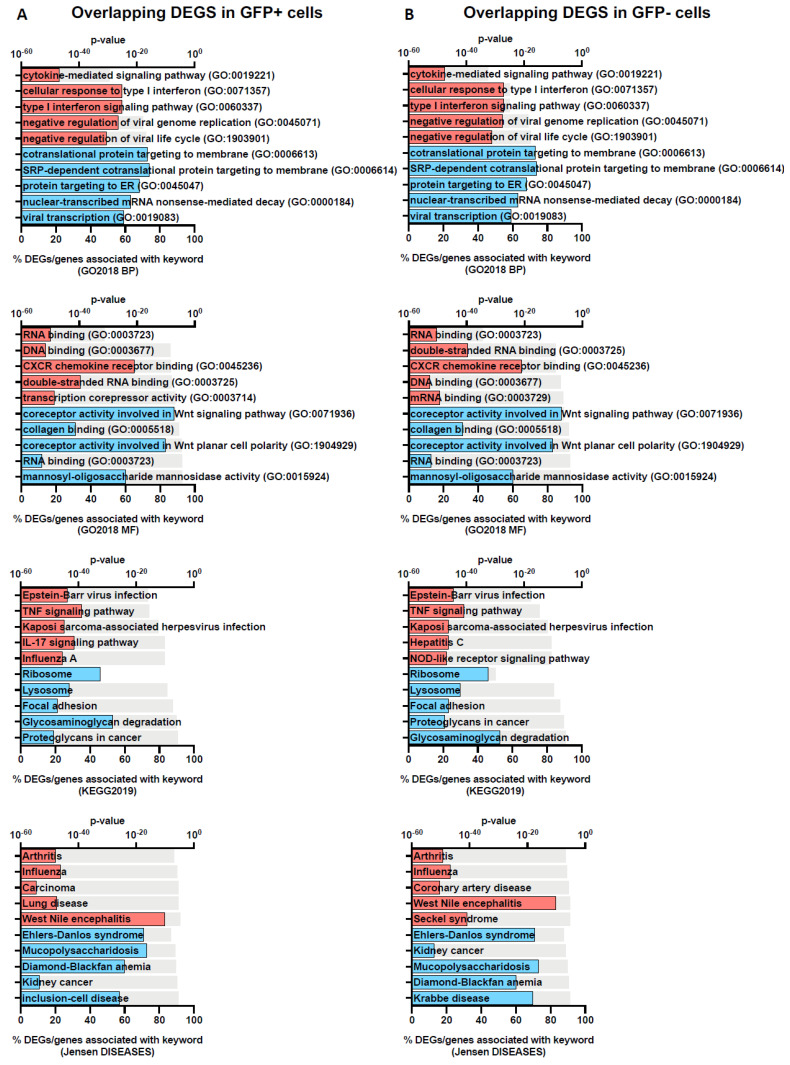
Gene-set enrichment analysis of overlapping DEGs in GFP+ and GFP− populations. The top 5 functional annotation for the two gene profiles: overlap-DEGs-GFP+ (**A**) and overlap-DEGs-GFP− (**B**) were identified using Enrichr with Gene Ontology Biological Process (GO2018 BP) and Molecular Function (GO2018 MF), KEGG and Jensen DISEASES databases. Keywords associated with up- and down-regulated genes are coloured red and blue, respectively. The grey bar represents the p-values determined by the Enrichr analysis platform.

**Figure 5 viruses-15-00136-f005:**
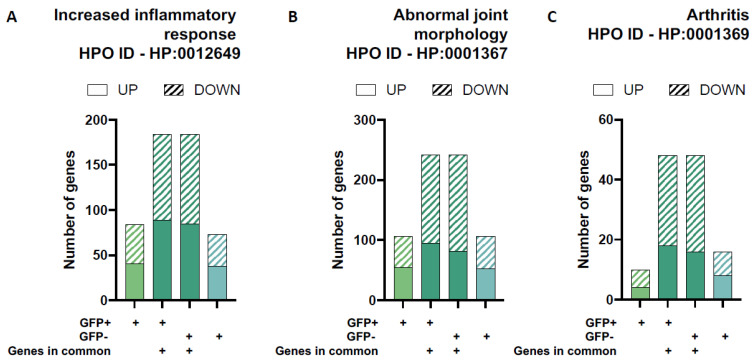
Differential gene expression for gene-sets related to joint pathology. The proportion of up- and down regulated differentially expressed genes (DEGs) in each of the four unique/overlapping gene profiles (GFP+ group only, overlapping genes in GFP+ cells, overlapping genes in GFP− cells and GFP− group only) were assessed using functional annotation gene-sets related to joint pathology from the Human Phenotype Ontology (HPO) database: increased inflammatory response (**A**), abnormal joint morphology (**B**) and arthritis (**C**).

**Figure 6 viruses-15-00136-f006:**
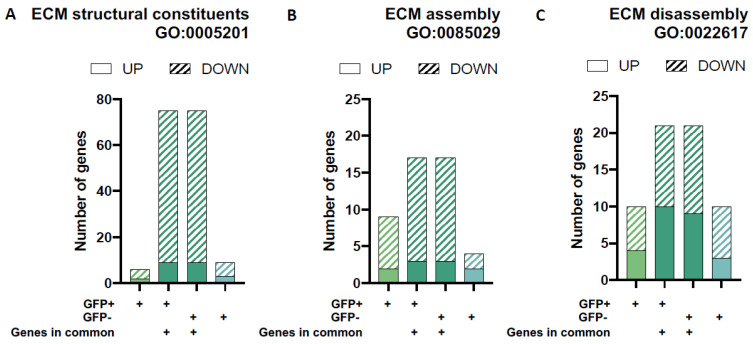
Differential gene expression for gene-sets associated with extracellular matrix (ECM) architecture and remodelling. The proportion of up- and down-regulated differentially expressed genes (DEGs) in each of the four unique/overlapping gene profiles (GFP+ group only, overlapping genes in GFP+ cells, overlapping genes in GFP− cells and GFP− group only) were assessed using functional annotation gene-sets related to extracellular matrix remodelling (ECM) from the Gene Ontology Biological Processes database (GO2018 BP): ECM structural constituents (**A**), ECM assembly (**B**) and ECM disassembly (**C**).

**Table 1 viruses-15-00136-t001:** Top ten up- and down-regulated differentially expressed genes in GFP+ cells (RRV-infected) based on their log2FC values.

	Relative Expression GFP+ Cells (Unique Genes)						Relative Expression GFP+ Cells (Overlapping Genes)				
	Symbol	Gene ID	log2FC	*p*-Value	FDR		Symbol	Gene ID	log2FC	*p*-Value	FDR
Up-regulated	*RIMS2*	9699	13.27	4.21E-07	1.33E-05		*DNAH2*	146754	15.87	9.07E-10	8.32E-07
	*PTGER3*	5733	12.75	1.06E-05	1.35E-04		*ACTN2*	88	15.32	4.91E-08	3.33E-06
	*FOXJ1*	2302	12.71	5.26E-07	1.60E-05		*HEATR9*	256957	15.23	3.23E-09	1.04E-06
	*LOC400499*	400499	12.71	4.22E-06	7.13E-05		*SCN3A*	6328	14.95	7.13E-09	1.34E-06
	*LRRC37A11P*	342666	12.53	1.78E-05	1.94E-04		*IL7R*	3575	14.93	4.90E-09	1.15E-06
	*INSM2*	84684	12.51	3.16E-06	5.77E-05		*NEFM*	4741	14.73	1.10E-06	2.70E-05
	*GUCY2D*	3000	12.43	1.21E-05	1.48E-04		*IFIT1B*	439996	14.65	1.26E-07	6.23E-06
	*GAS2L2*	246176	12.43	1.55E-04	9.98E-04		*MAFA*	389692	14.39	1.79E-07	7.77E-06
	*SLC9A2*	6549	12.37	3.42E-05	3.16E-04		*ARHGAP9*	64333	14.25	3.82E-09	1.09E-06
	*CATSPERD*	257062	12.31	2.81E-05	2.74E-04		*CXCL5*	6374	14.19	2.48E-10	8.32E-07

Down-regulated	*CGRRF1*	10668	−9.89	1.25E-03	5.04E-03		*GXYLT2*	727936	−11.22	1.21E-02	3.00E-02
	*MTM1*	4534	−9.74	1.75E-03	6.59E-03		*GCHFR*	2644	−11.14	5.39E-03	1.60E-02
	*MYPOP*	339344	−9.69	5.54E-03	1.63E-02		*MRPL48*	51642	−10.87	3.60E-04	1.91E-03
	*TCEANC2*	127428	−9.60	8.73E-04	3.79E-03		*KIT*	3815	−10.55	9.19E-03	2.43E-02
	*TMEM216*	51259	−9.54	2.26E-03	8.02E-03		*C22orf39*	128977	−10.54	4.03E-04	2.08E-03
	*ZNF415*	55786	−9.51	2.84E-03	9.64E-03		*MAP1LC3C*	440738	−10.36	1.63E-03	6.22E-03
	*PLAG1*	5324	−9.39	4.69E-03	1.42E-02		*EPHX2*	2053	−10.31	6.55E-04	3.02E-03
	*ARL6*	84100	−9.37	1.97E-03	7.23E-03		*FBLN7*	129804	−10.19	1.33E-03	5.30E-03
	*PPP1R3D*	5509	−9.36	2.17E-03	7.77E-03		*HOXA13*	3209	−10.03	1.80E-03	6.73E-03
	*FAM173B*	134145	−9.35	7.11E-03	1.99E-02		*CYP4X1*	260293	−9.97	1.77E-02	4.09E-02

**Table 2 viruses-15-00136-t002:** Top ten up- and down-regulated differentially expressed genes in GFP− cells (uninfected bystander) based on their log2FC values.

	Relative Expression GFP− Cells (Unique Genes)						Relative Expression GFP− Cells (Overlapping Genes)				
	Symbol	Gene ID	log2FC	*p*-Value	FDR		Symbol	Gene ID	log2FC	*p*-Value	FDR
Up-regulated	*GABBR2*	9568	11.02	1.94E-03	6.96E-03		*CXCL5*	6374	15.56	1.15E-10	9.09E-07
	*AIM2*	9447	8.46	3.81E-04	2.01E-03		*CXCL10*	3627	13.84	9.34E-07	3.17E-05
	*IFNL1*	282618	8.27	1.06E-04	7.87E-04		*CXCL11*	6373	13.79	8.81E-08	7.56E-06
	*CSF2*	1437	8.02	2.13E-04	1.30E-03		*IL7R*	3575	13.45	1.02E-08	2.26E-06
	*KRTAP2-3*	730755	8.00	2.16E-03	7.57E-03		*RSAD2*	91543	13.27	5.82E-10	9.09E-07
	*C3AR1*	719	7.92	3.93E-03	1.22E-02		*MMP1*	4312	12.78	1.41E-09	9.30E-07
	*CCL8*	6355	7.58	5.81E-04	2.74E-03		*ESM1*	11082	12.68	6.28E-08	6.46E-06
	*RNY4*	6086	6.36	2.01E-02	4.52E-02		*OASL*	8638	12.54	1.22E-09	9.09E-07
	*APOBEC3B*	9582	5.85	3.55E-04	1.90E-03		*BATF2*	116071	12.38	1.98E-08	3.15E-06
	*HIST1H3F*	8968	5.84	1.76E-04	1.13E-03		*GBP5*	115362	11.84	5.44E-07	2.20E-05

Down-regulated	*FRMD7*	90167	−9.26	1.30E-05	1.76E-04		*KRT4*	3851	−11.66	1.84E-07	1.17E-05
	*C1QTNF7*	114905	−8.85	7.21E-05	5.90E-04		*GDF10*	2662	−9.63	2.74E-05	2.97E-04
	*WSCD2*	9671	−8.74	1.09E-04	8.02E-04		*ATP1A2*	477	−8.34	2.83E-07	1.53E-05
	*CPAMD8*	27151	−7.75	5.38E-06	9.95E-05		*PRIMA1*	145270	−8.15	4.68E-07	2.00E-05
	*ALDH3A1*	218	−7.20	8.27E-06	1.29E-04		*ZBTB7C*	201501	−7.63	4.12E-07	1.86E-05
	*SCN2B*	6327	−6.37	4.61E-05	4.28E-04		*PTH1R*	5745	−7.33	2.38E-05	2.69E-04
	*HLF*	3131	−6.33	1.00E-05	1.47E-04		*YPEL1*	29799	−7.25	2.75E-05	2.98E-04
	*PRRT2*	112476	−6.30	7.39E-06	1.20E-04		*MAP1LC3C*	440738	−7.18	3.62E-06	7.67E-05
	*RASSF2*	9770	−6.25	1.06E-05	1.54E-04		*TENT5C*	54855	−6.60	1.72E-07	1.13E-05
	*REPS2*	9185	−6.10	4.11E-05	3.94E-04		*KLF15*	28999	−6.56	1.05E-08	2.28E-06

**Table 3 viruses-15-00136-t003:** Significant DEGs related to joint pathology and their association with alphavirus infection.

Symbol	Gene ID	Relative Expression GFP+ Cells				Relative Expression GFP− Cells				HPO (Increased Inflammatory Response)	HPO (Abnormal Joint Morphology)	HPO (Arthritis)	Associated with Alphavirus Infection
		↑/↓	log2FC	*p*-Value	FDR (Adjusted *p*-Value)	↑/↓	log2FC	*p*-Value	FDR (Adjusted *p*-Value)				
*ASPN*	54829	↓	−4.43	3.20E-06	5.84E-05	↓	−6.43	1.88E-08	3.15E-06		+	+	
*CA2*	760	↑	8.94	4.12E-06	7.01E-05	↑	5.25	1.50E-04	1.01E-03		+		
*EZH2*	2146	↑	6.19	3.47E-09	1.05E-06	↑	1.91	1.54E-05	1.97E-04		+		
*FOXJ1*	2302	↑	12.71	5.26E-07	1.60E-05	↑ *	2.82	8.07E-02	*1.38E-01*	+			
*GCH1*	2643	↑	5.87	3.30E-07	1.14E-05	↑	4.44	1.54E-06	4.41E-05		+	+	
*GORAB*	92344	↑	3.15	2.06E-07	8.34E-06	↑	0.68	6.65E-03	1.84E-02	+	+		
*HLA-B*	3106	↑	2.74	3.25E-07	1.13E-05	↑	3.22	6.46E-08	6.51E-06	+	+	+	[27]
*IFIH1*	64135	↑	7.32	7.49E-10	8.32E-07	↑	5.45	4.25E-09	1.28E-06	+	+	+	[28]
*IL6*	3569	↑	6.97	8.40E-08	4.70E-06	↑	6.35	1.21E-07	9.16E-06	+	+	+	[29]
*IL7R*	3575	↑	14.93	4.90E-09	1.15E-06	↑	13.45	1.02E-08	2.26E-06	+			[30]
*KDM6A*	7403	↑	3.62	5.06E-08	3.39E-06	↑	1.11	2.44E-04	1.43E-03	+	+		
*MAGI2*	9863	↑	3.63	3.81E-07	1.27E-05	↓	−1.45	1.82E-04	1.15E-03	+			
*MMP1*	4312	↑	11.49	1.15E-08	1.76E-06	↑	12.78	1.41E-09	9.30E-07	+			[31]
*MPDU1*	9526	↑	4.17	4.96E-07	1.53E-05	↑	1.45	6.07E-04	2.84E-03	+			
*MSX1*	4487	↑	5.32	9.80E-08	5.16E-06	↑	1.30	1.77E-03	6.50E-03	+			
*NFKB1*	4790	↑	3.85	1.11E-07	5.67E-06	↑	1.03	1.21E-03	4.87E-03	+			[32]
*NFKB2*	4791	↑	4.52	1.71E-08	2.12E-06	↑	1.74	1.58E-05	2.02E-04	+			[33]
*PNP*	4860	↑	4.03	1.11E-08	1.74E-06	↑	2.89	8.03E-08	7.25E-06	+			
*PRG4*	10216	↓	−3.35	5.25E-07	1.60E-05	↓	−3.73	4.23E-08	5.20E-06	+	+	+	
*RELB*	5971	↑	3.29	2.54E-07	9.68E-06	↑	2.66	4.54E-07	1.95E-05	+			[34]
*RIPK1*	8737	↑	4.37	2.16E-08	2.35E-06	↑	0.63	1.30E-02	3.16E-02	+			[35]
*SALL4*	57167	↑	10.92	1.65E-07	7.41E-06	↑ *	2.42	4.60E-02	*8.75E-02*		+		
*SAMD9*	54809	↑	3.82	2.46E-08	2.38E-06	↑	4.60	4.49E-09	1.28E-06	+			
*SERPINA1*	5265	↑	5.27	1.40E-07	6.67E-06	↑	5.80	3.71E-08	4.85E-06	+			
*SLC39A8*	64116	↑	4.41	2.31E-08	2.36E-06	↑	2.03	4.82E-06	9.23E-05		+		
*SLC40A1*	30061	↓	−4.95	2.11E-05	2.20E-04	↓	−6.02	1.79E-07	1.16E-05		+	+	
*TAP1*	6890	↑	3.26	4.03E-07	1.30E-05	↑	3.90	5.94E-08	6.19E-06	+			[14]
*TNFAIP3*	7128	↑	6.23	1.35E-09	8.33E-07	↑	5.30	3.22E-09	1.25E-06	+	+		[36]
*TNXB*	7148	↓	−1.77	2.22E-02	4.88E-02	↓	−4.71	1.28E-05	1.75E-04		+		
*VWF*	7450	↑	9.97	4.30E-08	3.17E-06	↑	2.10	1.45E-02	3.45E-02		+		
*WRAP53*	55135	↑	5.13	1.72E-07	7.56E-06	↑ *	0.68	5.37E-02	*9.94E-02*	+			

Upregulation and downregulation of gene expression are represented by ↑ and ↓ respectively. Values that do not meet the criteria for DEG in this study are annotated with an asterisk *. Genes associated with the categories in the table are listed with a “+” sign.

## Data Availability

Not applicable.

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
