# Peer review of "Pathways Activated by Infected and Bystander Chondrocytes in Response to Ross River Virus Infection"

_viruses, 2022, doi:10.3390/v15010136_

Round 1

Reviewer 1 Report

In this manuscript, the authors do an excellent job of conveying critical and unique transcriptomics signatures of RRV infection in chondrocytes. The experimental design is sound and the data are well presented. The compilation of the manuscript is rational and the discussion does a great job of identifying genes that are worthy of follow up studies. There is no doubt that this article will be of value to the alphavirus community as some genes do not appear to have been reported as modulated in the context of other alphavirus infections. The only suggestion for improvement is to include a replication-incompetent virus control, only as part of a small subset of genes that the authors would like to prioritize and independently query the expression changes by a targeted assay such as qPCR. This one additional experiment could also add a couple of lower MOIs of infection as the transcriptomics study was performed at a very high MOI. Such an experiment will not only show infectious dose dependency of the modulation in the target gene expression, but also demonstrate that it is a consequence of not just an infection, but ongoing virus replication. 

Author Response

Responses to reviewer’s comments:

We would like to express our sincerest gratitude to the Reviewers for the effort and time they have taken to provide constructive feedback on our manuscript. The Reviewers have made several suggestions and our team has done our best incorporate and accommodate these in the current version of the manuscript. We have presented the Reviewer’s comments along with our responses below. We attached a track changed version of the manuscript to easily review the additions/changes, in addition to the clean copy, as required.

Reviewer #1: The only suggestion for improvement is to include a replication-incompetent virus control, only as part of a small subset of genes that the authors would like to prioritize and independently query the expression changes by a targeted assay such as qPCR. Including MOIs.

We thank the reviewer for their suggestion. We can confirm that we optimised the infection and MOI prior to the study. We recognise the value in the reviewers suggestion of extra experiments, however that this would require months of experiments and analysis and is therefore beyond the scope of our study.  

Reviewer 2 Report

Reviewer report

In this study, the authors addressed the gene expression profiles of Ross River virus, an emerging Alphavirus of public health concern. The authors infected chondrocytes, that plays an important role in RRV disease pathogenesis, with a RRV isolate containing a GFP tag for two days. They then used RNA-sequencing (RNA-Seq), various software and FACS to identify differentially expressed genes (DEGs) expressed by RRV infected (GFP+) and bystander chondrocytes (GFP-). They reported various overlapping and unique DEGs expressed in the two cell groups. Using enrichment analysis, the authors further identified numerous DEGs related to joint pathology with no known association with alphavirus infection. They also identified genes related to inflammatory responses in both groups demonstrating immune responses regardless of infection status, an import aspect in therapeutic development.  

Overall, this is a well written manuscript with sound results from a well thought out and executed study. The manuscript provides interesting and novel findings, and a great steppingstone for proteomic and functional approaches that could lead to therapeutics.

Minor comment:

The introduction can be slightly shorter; the paragraph on pro-inflammatory responses can be moved to the discussion as at this stage the reader do not know that results will indicate pro-inflammatory responses as its not mentioned in the title or abstract. Rather discuss.

Methods:

The authors need to add more detail to the methods especially describing the T48 clone construct.

Results and discussion:

The first part of the results is not discussed. The discussion starts at table 3 with no indication of the significance of tale 1 and 2. Are there any literature on the DEGs from table 1 or 2? Could add that. Please elaborate in the text (methods) or title indicating how the top ten was selected and arranged.

For table 1 and 2, although DEGS are overlapping between GPF+ and GFP – cells, the top 10 indicated in table 1 and 2 are not 100% identical. Meaning that there are differences in the overlapping expression profiles between the two groups. The authors might like to add a sentence addressing this especially for future proteomic work.  

Figures

·         Please ensure fonts are the same throughout the figures.

·         Elaborate more in the figure legends. One should be able to understand the figure/ table without going back to the text in the results section.

·         Figure 1’s labels (blue/red dots) are unreadable. Please increase the font size.

·         Figure 1’s legend should be moved up and the text in lines 153-160 should be moved down so that the legend is directly below the figure.

Specific comments:

Line 31. Add “by RRV”: “…responses elicited by RRV in cells…”

Line 69. Add “if/whether” :..disease pathogenesis or if/whether the death…”

Line 78-84. Very long sentence. Could split after “…viral-induced arthritis. Some of these soluble factors have also been found...”

Line 102: Which growth media was used?

Line 102: Please reference or describe the T48 clone construct

Line 205: Should be table 3 not Table 1

Line 214: add "respectively" to clarify both groups have unique sets: (≃ 5 unique genes respectively)”

Line 215: to many space after “)” & “and”

Line 238: Specify what HP:0012649 etc means. eg. (A, HPO ID= HP:0012649)

Line 216-263: Check sentence tense. Might need to change “encode” to “encoding”

Line 278: Add reference

Line 353: Add “is”: “SALL4 expression is virtually…”

Please remove extra spaces in reference list e.g. line 401

Author Response

Reviewer #2: Minor comment: 

The introduction can be slightly shorter; the paragraph on pro-inflammatory responses can be moved to the discussion as at this stage the reader do not know that results will indicate pro-inflammatory responses as its not mentioned in the title or abstract. Rather discuss.

Methods:

The authors need to add more detail to the methods especially describing the T48 clone construct. 

We have now added extra details to describe the methods, including details and references for the T48 infection clone. Please see the revised manuscript.

Results and discussion: 

The first part of the results is not discussed. The discussion starts at table 3 with no indication of the significance of tale 1 and 2. Are there any literature on the DEGs from table 1 or 2? Could add that. Please elaborate in the text (methods) or title indicating how the top ten was selected and arranged. 

These changes have now been made with additions to the discussion and methods

For table 1 and 2, although DEGS are overlapping between GPF+ and GFP – cells, the top 10 indicated in table 1 and 2 are not 100% identical. Meaning that there are differences in the overlapping expression profiles between the two groups. The authors might like to add a sentence addressing this especially for future proteomic work.  

This has now been added

Figures

Please ensure fonts are the same throughout the figures. 

Corrected

Elaborate more in the figure legends. One should be able to understand the figure/ table without going back to the text in the results section.

Additions have been made to the figure legends in line with the reviewer suggestions

      Figure 1’s labels (blue/red dots) are unreadable. Please increase the font size.

The font size has been increased

Figure 1’s legend should be moved up and the text in lines 153-160 should be moved down so that  the legend is directly below the figure. 

This has been moved

Specific comments:

Line 31. Add “by RRV”: “…responses elicited by RRV in cells…”

This has now been added

Line 69. Add “if/whether” :..disease pathogenesis or if/whether the death…”

This has now been added

Line 78-84. Very long sentence. Could split after “…viral-induced arthritis. Some of these soluble factors have also been found...”

 This sentence has been split into two as suggested by the reviewer

Line 102: Which growth media was used? Line 102: Please reference or describe the T48 clone construct

Two new sections have been added to the methods to better describe the cell culture techniques, media, and the infectious clone. This includes additional references.

Line 205: Should be table 3 not Table 1

This has been corrected

Line 214: add "respectively" to clarify both groups have unique sets: (≃ 5 unique genes respectively)”

This has been added

Line 215: to many space after “)” & “and”

This has been corrected

Line 238: Specify what HP:0012649 etc means. eg. (A, HPO ID= HP:0012649)

This has been clarified

Line 216-263: Check sentence tense. Might need to change “encode” to “encoding”

This has been corrected

Line 278: Add reference

This has been added

Line 353: Add “is”: “SALL4 expression is virtually…”

This has been added

Please remove extra spaces in reference list e.g. line 401

This has been corrected
